# Temporal Interactions between Root-Lesion Nematodes and the Fungus *Rhizoctonia Solani* Lead to Reduced Potato Yield

**Eva Edin** [1,*] ⓘ**, Mehreen Gulsher** [2]**, Mikael Andersson Franko** [3]**, Jan-Eric Englund** [4]**, Adam Flöhr** [4]**, Jonas Kardell** [2] **and Maria Viketoft** [2]

[1]   Department of Forest Mycology and Plant Pathology, Swedish University of Agricultural Sciences (SLU),
      PO Box 7026, 750 07 Uppsala, Sweden
[2]   Department of Ecology, Swedish University of Agricultural Sciences (SLU), PO Box 7044,
      750 07 Uppsala, Sweden
[3]   Department of Energy and Technology, Swedish University of Agricultural Sciences (SLU), PO Box 7032,
      750 07 Uppsala, Sweden
[4]   Department of Biosystems and Technology, Swedish University of Agricultural Sciences (SLU), PO Box 103,
      230 53 Alnarp, Sweden
*   Correspondence: eva.edin@slu.se; Tel.: +46-18671000; Fax.: +46-18673599

**Abstract:** Soil microorganisms and soil fauna may have a large impact on the tuber yield of potato crops. The interaction between root-lesion nematodes and the pathogenic fungus *Rhizoctonia solani* Kühn was studied on potato plants grown in pots under controlled conditions. In two similar experiments, different combinations of nematodes and fungal mycelium were added to the pots at three occasions; at planting, after 14 days, and after 28 days. The nematodes reduced root biomass and the combination of nematodes and *R. solani* resulted in reduced tuber yield in both experiments, but the interaction was not synergistic. In contrast, the number of stem canker lesions decreased in the presence of nematodes compared to treatments with *R. solani* only. The time of inoculation influenced the severity of both fungal and nematode damage. The nematode damage on tubers was less severe if the nematodes were added at 28 days, while the number of severe stem canker lesions increased if the fungus was added at 28 days. However, the time of nematode inoculation did not affect the incidence of fungal damage, hence the nematodes did not assist *R. solani* to infect the plant. Our results highlight the underestimated importance of root-lesion nematodes, not resulting in obvious above ground symptoms or misshaped tubers yet affecting the performance of other pathogens.

**Keywords:** black scurf; disease complex; elephant hide; *Pratylenchus*; stem canker

## 1. Introduction

Producing a high-yielding and high-quality potato crop (*Solanum tuberosum* L.) is a complex task with many aspects to consider, including cultivar, soil parameters, rainfall, and agricultural measures [1,2]. In addition, the organisms in the soil, such as fungal species, insects, and bacteria, may very well influence the development of the potato crop depending on their food preferences [3]. The complexity of the interaction between multiple species present in the soil and the cultivated crop is difficult to entangle [4]. Here, we investigated the temporal interactions between the potato plant root-lesion nematodes and the pathogenic fungus *Rhizoctonia solani* Kühn.

Free-living and sedentary plant-parasitic nematodes, representing 15% of the total number of nematode species described, are present in soils worldwide and are significant pathogens in agriculture [5]. Free-living plant-parasitic nematodes have a vast host range, including potato, and are

mobile during their entire life cycle, either in the soil (ectoparasites) or in the roots (endoparasites). The contribution of free-living plant-parasitic nematodes to yield losses is probably underestimated because aboveground symptoms or misshaped tubers seldom are prominent and may just delay the development of the plant. Root-lesion nematodes (endoparasitic free-living nematodes) may reduce tuber yield by 12% and even more if they are followed by secondary infections by fungi and bacteria [6]. Plant-parasitic nematodes may also affect the development of diseases caused by other soil-borne pathogens by enhancing the impact of a pathogenic fungus in a synergistic interaction [7]. For example, nematodes may react to the exudates from the fungal infected plants or the fungus may penetrate the plant cortex through the wounds caused by nematode feeding [7].

The fungal pathogen *R. solani* causes stem canker, black scurf, or deformed tubers, among other symptoms like elephant hide and skin cracks on potato plants [8,9]. The fungus can survive on harvest residues in the soil or be seed-borne when it is present on the seed tubers as sclerotia (black scurf). Soil-borne inoculum is more troublesome, and plant-parasitic nematodes have been shown to be more abundant in patches of stem canker, caused by soil-borne *R. solani*, in commercial Swedish potato fields [10]. Interaction effects between *R. solani* and free-living nematodes on potato tuber yield are not so well studied. The root-lesion nematode *Pratylenchus penetrans* (Cobb, 1917) has been reported to influence the effect of *R. solani* but did not cause any yield reduction [11]. Instead, the severity of potato early dying symptoms caused by *Verticillium dalhiae* Kleb. increased in co-occurrence with *P. penetrans* and *R. solani*. A previous study revealed that a full nematode community, dominated by the root-lesion nematode *Pratylenchus crenatus* (Loof, 1960), had a negative impact on tuber yield in a pot experiment [12]. Other nematode species interacting with *R. solani* are the potato cyst nematodes *Globodera rostochiensis* (Wollenweber, 1923) [13,14] and *Globodera pallida* (Stone, 1973), 1975 [15,16], which can result in large yield losses. In soybean, high densities of root-lesion nematodes (*Pratylenchus* spp.) and stunt nematodes (*Tylenchorhynchus* spp.) have been observed in field patches with severe disease symptoms caused by *R. solani* [17].

The exposure time of the plant to the pathogen and the nematode, as well as the development stage of the host plant, may be important for the interaction between the organisms and disease severity for the host [18]. The temporal aspects of exposure and susceptibility due to plant development have not been sufficiently studied, even though there are studies that compare cultivar susceptibility [19,20].

The aim of the present study was to determine if the time of introduction of root-lesion nematodes and the pathogenic fungus *R. solani* on potato plants has an impact on the severity of the subsequent symptoms and yield loss. The experiment was performed twice, first with a full nematode community, containing a mixture of plant-parasitic nematodes dominated by root-lesion nematodes and non-parasitic nematodes, and secondly with the root-lesion nematode *P. penetrans*. The potato plants were grown in pots under controlled conditions with the pathogens added at three-time points in different combinations. The hypotheses were that the combination of plant-parasitic nematodes and *R. solani* would result in (1) more severe canker on stems and stolons, (2) reduced quality of the tubers due to high incidence of black scurf and other tuber symptoms, and (3) concomitant lower tuber yield. We also hypothesized that the time of inoculation would (4) affect the number of fungal symptoms and nematode damage and (5) nematode inoculation prior to fungal inoculation would give a stronger interaction.

## 2. Materials and Methods

This study consists of two separate experiments with some differences in the experimental setup and the nematodes used. With reference to observations from experiment 1 and the experimental set up, we performed a repetition of the experiment with all treatment combinations for species and time of inoculation as in experiment 2.

### 2.1. Fungus

An isolate of *R. solani*, AG2-1, originating from Vara, Sweden, was kindly provided by Dr. S. Ahlström (Dept. of Forest Mycology and Plant Pathology SLU, Uppsala, Sweden), which was the same isolate as that used in the experiments previously published [12,21]. The methodology was also the same as in those experiments, that is, the fungal mycelium was mixed to small pieces and diluted in tap water to correspond to 0.01 g mycelium pot$^{-1}$ within the same volume as for the inoculation of the nematodes.

### 2.2. Nematodes

Two types of nematode inocula were used for the experiments; in experiment 1, a full nematode community dominated by root-lesion nematodes, in particular *P. crenatus*, and in experiment 2, a pure culture of the root-lesion nematode *P. penetrans*. The full nematode community was derived from field soil taken from a potato field in the county of Östergötland in south-central Sweden, whereas *P. penetrans* were originally bought from Plant Research International Wageningen, the Netherlands, and kept in culture on maize. The maize plants were grown in sterilized sand and new seeds were sown after the senescence of the plants, which were cut, and the roots were left in the pots as feed for the nematodes. The nematodes were extracted from soil and root tissues, respectively, using Whitehead and Hemming trays (Whitehead and Hemming 1965), kept in a cold storage room 4 °C, and aerated with an aquarium air pump (superfish Air-Flow mini). The concentration of plant-parasitic nematodes in the solution was determined by counting the nematode numbers in subsamples under the microscope.

### 2.3. Experimental Setup

The experiments were designed as pot experiments with eight and sixteen treatments, respectively, (Table 1) with eight replicates. Washed and sterilized (200 °C for six hours) sand (0–3 mm Ø) was used as potting medium. In each pot ($12 \times 12 \times 25$ cm), one filter paper was used to cover the bottom to avoid leakage of the 1600 g dry sand that was first added. The potting medium was then wetted with 200 ml of tap water. One pre-germinated mini tuber of cultivar King Edward VII (tubers produced from meristem cultures) (Agrico Nordic, cultivated at The Finnish Seed Potato Centre Ltd (SPK), Tyrnävä, Finland) was placed in each pot with the most germinated sprouts facing upwards. Another 900 g of sand was added on top of the tuber and 100 ml tap water was applied.

For both experiments, the first inoculation of organisms occurred the day after planting. Nematodes, fungus, and water were added in different combinations to the pots: control (tap water); fungus (0.1 g mycelium pot$^{-1}$); nematodes (2 plant-parasitic nematodes g$^{-1}$ sand) at start, after 14 days, and after 28 days. During the inoculation of nematodes in experiment 1, three subsamples were taken out to determine the complete composition of the nematode community added to the pots. The actual inoculum volume differed between the inoculation events as a result of variations in nematode density, but the volumes added for each treatment were always the same within each inoculation time. For instance, at one inoculation event, the volume of 90 ml of liquid solution contained either 45 ml of nematode solution, 45 ml fungal solution, and/or tap water, according to the treatments in Table 1. The remaining pots received the corresponding amount of water at each inoculation event.

The experiments were carried out in a climate chamber set to 12 °C and with a day/night cycle of 16/8 h of artificial light (152 LUX on average). The pots were placed on trolleys in a randomized complete block design and the trolleys were moved once a week to provide a uniform exposure to light. The plants were initially watered (200 mL) twice a week to ensure normal moisture and during the last weeks of the experiments, depending on the growth level, the plants were watered thrice a week. In addition, all pots were fertilized at three occasions in experiment 1 and four occasions in experiment 2 with a complete fertilizer (Blomstra NPK: 100:18:86 and micronutrients, Orkla Care, Solna, Sweden). In total, each plant received 180 mg nitrogen and 200 mg nitrogen, respectively.

**Table 1.** Experimental setup for inoculations in first and second experiment, including abbreviations of the treatment combinations. Treatments included in experiment 1 with full nematode community are denoted with *.

| | Nematode Addition (N) | | | |
|---|---|---|---|---|
| **Fungus (F)** | **Not added** | **Start** | **Day 14** | **Day 28** |
| **Not added** | Control * | N * | N14 | N28 |
| **Start** | F * | N + F * | F + N14 * | F + N28 * |
| **Day 14** | F14 | N + F14 * | F14 + N14 | F14 + N28 |
| **Day 28** | F28 | N + F28 * | N14 + F28 | N28 + F28 |

*2.4. Harvest*

The potato plants were harvested ten weeks after the first inoculation. This was performed as previously described [12,21]. In short, the potato plant was taken out of the pot; washed carefully with tap water; dried with a paper towel; and divided into stems, roots, stolons, and tubers. The tubers were divided into small (0.5–2 cm in diameter) and large (>2 cm in diameter) tubers. Tubers smaller than 0.5 cm were regarded as stolons. In experiment 1, the plant parts were graded regarding nematode damage, stem canker, sclerotia, and "elephant hide" (Table A1). The severity of stem canker and "elephant hide" were graded using the same method as in experiment 2, while the actual number of sclerotia and nematode damage was instead counted on each plant part.

For the treatments with nematodes, the submerged part of the main stem (from the mother tuber up to the surface level, 5–8 cm), the largest tuber, approximately ten roots, and 20 g of the potting medium were weighed separately and put in plastic bags for cold storage until later extraction of nematodes. The remaining plant parts and 100 g of the potting medium were weighed and dried for dry weight measurements. The mother tuber was measured in length and diameter and graded regarding nematode damage and sclerotia (black scurf).

*2.5. Nematode Extraction*

The below ground part of the main stem and the roots were cut into 1 cm pieces and each put in one vlieseline covered mesh net sieve. The tuber was divided into four pieces and put skin side down in each mesh net sieve or, if too large, two sieves per tuber were used. The entire 20 g sample of potting sand was placed in a mesh net sieve. The sieves were placed in Baermann funnels, and the nematodes were extracted for 24 h, heat-killed, and fixated in formalin [22]. The number of nematodes was estimated in the suspensions from each extraction under low magnification (50×) and expressed as the number of nematodes per gram dry weight of each plant part or per gram dry potting sand.

*2.6. Statistical Analyses*

The two experiments were analysed separately.

In experiment 1, the effects of additions of fungus and nematodes on dry weight of stems, tubers, roots, and stolons were analysed with a randomized complete block design analysis of variance (ANOVA) using R version 3.1.1 [23]. To account for extreme values and heteroscedasticity, we used robust standard errors, packages multcomp [1], and sandwich [2,3]. The number of tubers and stems was analysed with Poisson regression using R, with the control treatment used as baseline. The proportion of small tubers was analysed with binomial regression using SAS for Windows 9.3 (SAS Institute Inc., Cary, NC, USA). To analyse the effect of fungal and nematode damage, we used a dichotomous variable-damage or no damage. The data were analysed with binomial regression using R. The number of nematodes in the different plant parts and in the sand was analysed with randomized complete block design ANOVA in R with robust standard errors.

For experiment 2, stem, root, and stolon biomass; number of tubers; and abundance of root-lesion nematodes in roots were analyzed with linear mixed models with block as a random factor using

R version 3.5.2 [23] and packages lme4 [24] and nlme [25]. The log-transformed biomass was used for stems and stolons as a result of better results in tests for normality and equal variance for the residuals (diagnostics test performed using the car package [26]). Abundance of root-lesion nematodes in roots was modelled using log-transformed abundance and a treatment-dependent variance, thanks to differences in variance.

The probability of elephant hide was modelled with a logistic model. The factor of nematode addition was dropped thanks to non-significance in the model with interaction. Block and treatment interacting with block were used as random factors. Stem canker was modelled with a negative-binomial model on number of lesions. Separate models were estimated for lesions of severity 1 and for severity 2 and 3 combined. Block and treatment nested within block were used as random factors.

Post-hoc comparisons between factor levels were done using the Tukey honestly significant difference (HSD) method in the emmeans package [27]. For all models in experiment 2, model selection was performed through backward elimination based on significance in Type II Wald-tests. The interaction term was tested by comparing the full-factorial model to the additive model. Non-significant factors were first reduced to binary factors (added or not added nematodes or fungi respectively) and, if that binary factor was non-significant, removed from the model.

## 3. Results

The potato plants developed well and the shoots were approximately two cm above the surface of the potting medium at the time of the second inoculation and all, except two tubers in experiment 2, had germinated. The plants were approximately 10 cm at the inoculation at day 28 after planting. If the potato plant was inoculated with *R. solani*, it was clearly affected and had symptoms like stem canker, black scurf, and elephant hide. Plant-parasitic nematodes were present in all tested plant parts as well as in the potting medium from the pots where nematodes had been added. In the first experiment with a full nematode community, plant-parasitic nematodes dominated the inoculated community together with bacterivorous nematodes (47% each) and consisted of the taxa *Pratylenchus* (41% of total nematode abundance), *Tylenchorhynchus* (5%), and Trichodoridae (1.4%). As expected, only the endoparasitic *Pratylenchus* spp. were found in the different plant parts at harvest. The nematode community also consisted of fungal-feeding nematodes from the genera *Aphelenchus* (1.4%) and *Aphelenchoides* (2.9%), and some omnivors (1.4%) were also present.

### 3.1. Impact on Stems

The dry weight and number of stems were not significantly affected by treatment in either of the experiments (E1: $p = 0.31$ and $p = 0.34$, respectively; E2: $p = 0.52$ and $p = 0.39$, respectively; Table 2). The majority of the stems got stem canker in both experiments regardless of the time that *R. solani* was added (Tables 3 and 4). Nematode addition reduced the severity of stem canker in experiment 2, as the number of stem canker lesions of severity 1 (small lesions) was reduced to 2.81 lesions stem$^{-1}$ in the treatments that also contained nematodes, compared with 4.15 lesions stem$^{-1}$ without nematode addition ($p = 0.0024$). Likewise, the number of lesions with severity 2 (large lesion) or 3 (completely girdled) increased with 82% in absence of nematodes ($p = 0.0096$). The occurrence of the fungus did not affect the number of nematodes in the stems in either of the experiments ($p = 0.54$ and $p = 0.12$, respectively; Table 5).

The time of fungal inoculation influenced the number of stem canker lesions, as the likelihood of severe stem canker was lower in the treatment with nematode addition at planting and fungus at 28 days in experiment 1 ($p = 0.029$). In addition, in experiment 2, the number of stem canker lesions of severity 2 and 3 was 1.94 times higher when the fungus was inoculated at 28 days compared with inoculation at 14 days ($p = 0.027$). The nematode damage in experiment 2, as well as the number of nematodes per gram stem, were significantly higher if the nematodes were added at the start and at 14 days compared with at 28 days ($p < 0.001$; Table 6).

**Table 2.** Dry weights (g; mean (SE)) and numbers of different parts of potato plants subjected to different combinations of the plant pathogenic fungus *Rhizoctonia solani* (F) and nematodes (N)—full community in experiment 1 and the root-lesion nematode *Pratylenchus penetrans* in experiment 2. Treatments with the same letter within a column and experiment are not significantly different at level $p < 0.05$.

| Treatment | No. of Stems | Stem Biomass | Stolon Biomass | Root Biomass | No. of Tubers |
|---|---|---|---|---|---|
| **Experiment 1** | | | | | |
| C | 1.38 (0.17) | 1.99 (0.08) | 0.19 (0.02) a | 0.35 (0.01) | 4.00 (0.35) |
| F | 1.88 (0.28) | 1.87 (0.15) | 0.05 (0.11) b | 0.35 (0.03) | 4.25 (0.52) |
| N | 1.63 (0.35) | 9.94 (0.01) | 0.18 (0.02) a | 0.33 (0.02) | 4.50 (0.64) |
| N + F | 1.25 (0.15) | 1.80 (0.07) | 0.05 (0.01) b | 0.31 (0.03) | 4.13 (0.87) |
| N + F14 | 1.38 (0.17) | 1.55 (0.24) | 0.03 (0.01) b | 0.24 (0.05) | 4.13 (0.67) |
| N + F28 | 1.75 (0.34) | 1.98 (0.17) | 0.16 (0.03) a | 0.41 (0.04) | 4.00 (0.56) |
| F + N14 | 2.63 (0.30) | 1.83 (0.12) | 0.06 (0.01) b | 0.31 (0.03) | 5.75 (0.84) |
| F + N28 | 1.88 (0.33) | 1.74 (0.09) | 0.05 (0.01) b | 0.31 (0.02) | 4.00 (0.43) |
| **Experiment 2** | | | | | |
| C | 1.63 (0.2) | 4.52 (0.2) | 0.21 (0.03) ab | 0.84 (0.09) ab | 6.50 (0.6) |
| F | 1.50 (0.3) | 3.95 (0.2) | 0.09 (0.02) abc | 0.78 (0.08) ab | 4.63 (0.6) |
| F14 | 2.00 (0.3) | 4.49 (0.3) | 0.11 (0.06) abc | 0.80 (0.06) ab | 5.29 (1.0) |
| F28 | 2.13 (0.1) | 4.20 (0.3) | 0.15 (0.03) abc | 0.90 (0.08) a | 7.00 (1.1) |
| N | 2.13 (0.3) | 4.00 (0.2) | 0.16 (0.03) abc | 0.65 (0.06) ab | 7.25 (1.1) |
| N14 | 2.13 (0.1) | 4.05 (0.2) | 0.16 (0.03) abc | 0.61 (0.05) ab | 8.25 (0.6) |
| N28 | 1.88 (0.3) | 4.14 (0.2) | 0.23 (0.02) a | 0.80 (0.08) ab | 6.88 (0.5) |
| N + F | 2.00 (0.3) | 4.13 (0.4) | 0.10 (0.03) abc | 0.57 (0.05) b | 8.50 (1.4) |
| N + F14 | 2.00 (0.0) | 4.32 (0.2) | 0.08 (0.03) bc | 0.73 (0.07) ab | 5.88 (0.6) |
| N + F28 | 2.29 (0.3) | 3.86 (0.3) | 0.17 (0.02) abc | 0.60 (0.08) ab | 5.29 (1.2) |
| F + N14 | 2.38 (0.3) | 4.24 (0.2) | 0.10 (0.02) abc | 0.71 (0.07) ab | 6.75 (1.1) |
| F + N28 | 1.75 (0.2) | 3.95 (0.1) | 0.13 (0.02) abc | 0.60 (0.06) ab | 6.63 (0.8) |
| F14 + N14 | 1.75 (0.2) | 3.94 (0.1) | 0.06 (0.02) c | 0.67 (0.06) ab | 7.25 (1.4) |
| F14 + N28 | 1.88 (0.2) | 4.03 (0.3) | 0.09 (0.03) c | 0.55 (0.06) b | 7.00 (1.4) |
| N14 + F28 | 1.88 (0.1) | 3.76 (0.2) | 0.14 (0.02) abc | 0.64 (0.04) ab | 6.00 (1.2) |
| N28 + F28 | 2.38 (0.3) | 4.18 (0.1) | 0.14 (0.02) abc | 0.80 (0.07) ab | 7.13 (1.1) |

**Table 3.** Percentage d amaged parts of potato plants subjected to the fungus *Rhizoctonia solani* (F) and full nematode community (N) in experiment 1. Treatments with the same letter within a column and experiment are not significantly different at a level of $p < 0.05$.

| Treatment | Stem | | Tubers | | | Stolons [I] | | | | Roots | | |
|---|---|---|---|---|---|---|---|---|---|---|---|---|
| | Stem Canker | Sclerotia | Nematode Damage | Elephant Hide | Black Scurf | Nematode Damage | Sclerotia | Lesions | % Brown Stolons | Sclerotia | Lesions | % Brown Roots |
| **Experiment 1** | | | | | | | | | | | | |
| C | 0 [II] | 0 | 0 | 0 | 0 | 0 | 0 | 0 | 25 | 0 | 0 | 100 [III] |
| F | 87 | 40 | 0 | 32 b | 68 ab | 0 | 25 | 0 | 75 | 88 | 0 | 100 |
| N | 0 | 15 | 92 | 0 | 0 | 0 | 0 | 88 | 38 | 0 | 63 | 100 |
| N + F | 90 | 80 | 100 | 45 ab | 42 ab | 0 | 25 | 50 | 88 | 100 | 100 | 100 |
| N + F14 | 100 | 82 | 64 | 39 ab | 42 ab | 0 | 38 | 63 | 63 | 75 | 63 | 88 |
| N + F28 | 64 | 57 | 79 | 78 a | 81 a | 0 | 75 | 75 | 100 | 88 | 100 | 100 |
| F + N14 | 100 | 57 | 71 | 39 b | 39 b | 0 | 13 | 25 | 75 | 100 | 75 | 100 |
| F + N28 | 67 | 40 | 60 | 34 b | 56 ab | 0 | 0 | 50 | 100 | 75 | 25 | 100 |

[I] Stolons were not recovered from every replicate in experiment 1. [II] percentage of all stem canker lesions, experiment 1. [III] percentage of root replicates with brown discoloration, experiment 1.

**Table 4.** Percentage of stems and tubers with fungal and nematode damage from potato plants subjected to different combinations of the plant pathogenic fungus *Rhizoctonia solani* (F) and the root-lesion nematode *Pratylenchus penetrans* (N) in experiment 2. Results from statistical analyses for experiment 2 are presented in Tables 6 and 7.

| Treatment | Stem | | | Tubers | | | Stolons [a] | | | | Roots | | |
|---|---|---|---|---|---|---|---|---|---|---|---|---|---|
| | Stem Canker | | Sclerotia | Nematode Damage | Elephant Hide | Black scurf | Nematode Damage | Sclerotia | Lesions | % Brown Stolons | Sclerotia | Lesions | % Brown Roots |
| **Experiment 2** | | | | | | | | | | | | | |
| C | 0 [I] | 0 [II] | 0 | 0 | 0 | 0 | 0 | 0 | 0 | 7 | 0 | 0 | 18 [III] |
| F | 77 | 69 | 77 | 0 | 54 | 51 | 0 | 50 | 0 | 24 | 75 | 0 | 23 |
| F14 | 86 | 64 | 93 | 0 | 78 | 76 | 0 | 86 | 0 | 32 | 100 | 0 | 24 |
| F28 | 88 | 59 | 94 | 0 | 70 | 84 | 0 | 100 | 0 | 42 | 100 | 0 | 36 |
| N | 0 | 0 | 0 | 100 | 0 | 0 | 33 | 0 | 100 | 11 | 0 | 100 | 34 |
| N14 | 0 | 0 | 0 | 100 | 0 | 0 | 44 | 0 | 100 | 5 | 0 | 100 | 36 |
| N28 | 0 | 0 | 0 | 73 | 0 | 0 | 20 | 0 | 100 | 6 | 0 | 100 | 27 |
| N + F | 75 | 69 | 88 | 94 | 37 | 62 | 41 | 88 | 75 | 30 | 100 | 100 | 36 |
| N + F14 | 88 | 63 | 94 | 100 | 66 | 66 | 36 | 100 | 100 | 30 | 100 | 100 | 31 |
| N + F28 | 93 | 60 | 100 | 100 | 70 | 68 | 32 | 100 | 100 | 29 | 100 | 100 | 29 |
| F + N14 | 84 | 58 | 79 | 100 | 85 | 67 | 35 | 75 | 100 | 31 | 100 | 100 | 31 |
| F + N28 | 57 | 36 | 43 | 86 | 19 | 26 | 9 | 38 | 100 | 17 | 38 | 100 | 29 |
| F14 + N14 | 86 | 50 | 93 | 100 | 72 | 74 | 45 | 63 | 88 | 33 | 63 | 100 | 34 |
| F14 + N28 | 87 | 67 | 73 | 100 | 75 | 80 | 21 | 38 | 100 | 31 | 38 | 100 | 31 |
| N14 + F28 | 87 | 53 | 80 | 100 | 60 | 77 | 27 | 88 | 100 | 34 | 88 | 100 | 36 |
| N28 + F28 | 84 | 58 | 89 | 89 | 54 | 58 | 5 | 100 | 100 | 28 | 100 | 100 | 31 |

[I] lesion of severity grade 1, experiment 2. [II] lesions of severity grade 2 and 3, experiment 2. [III] average percentage of brown coloration as graded on replicates within treatment, experiment 2.

**Table 5.** Abundances of plant-parasitic nematodes (PPN) and fungal-feeding nematodes (FFN) (no. gram$^{-1}$ dry weight; mean (SE)) in different parts of potato plants subjected to the plant pathogenic fungus *Rhizoctonia solani* (F) and nematodes (N)—full nematode community (experiment 1) and the root-lesion nematode *Pratylenchus penetrans* (experiment 2). Treatments with the same letter within a column for experiment 1 are not significantly different at a level of *p* < 0.05. Results from statistical analyses for experiment 2 are presented in Tables 6 and 7.

| Treatment | No. in Stems | | No. in Tubers | | No. in Roots | | No. in Potting Medium | |
|---|---|---|---|---|---|---|---|---|
| | PPN | FFN | PPN | FFN | PPN | FFN | PPN | FFN |
| **Experiment 1** | | | | | | | | |
| N | 31.1 (14.3) | 0.0 (0.0) b | 0.00 (0.00) | 0.00 (0.00) | 746.5 (283.1) | 0.0 (0.0) b | 0.05 (0.02) | 0.00 (0.00) |
| N + F | 25.6 (7.3) | 28.9 (16.0) ab | 0.02 (0.02) | 0.00 (0.00) | 419.9 (190.7) | 157.9 (103.7) ab | 0.01 (0.01) | 0.15 (0.05) |
| N + F14 | 67.2 (38.1) | 64.5 (15.6) a | 0.20 (0.10) | 0.06 (0.05) | 756.1 (232.1) | 91.5 (42.5) b | 0.04 (0.02) | 0.28 (0.16) |
| N + F28 | 26.6 (15.1) | 10.5 (3.0) b | 0.13 (0.06) | 0.02 (0.02) | 580.7 (155.4) | 32.8 (10.5) b | 0.05 (0.02) | 0.03 (0.01) |
| F + N14 | 54.0 (20.3) | 29.4 (17.0) ab | 0.05 (0.03) | 0.03 (0.03) | 473.5 (81.0) | 472.9 (124.7) a | 0.06 (0.03) | 0.47 (0.20) |
| F + N28 | 20.6 (6.0) | 64.1 (20.3) a | 0.19 (0.06) | 0.04 (0.04) | 175.1 (53.3) | 245.4 (99.6) ab | 0.01 (0.01) | 0.11 (0.02) |
| **Experiment 2** | | | | | | | | |
| N | 122.1 (32.3) | - | 1.63 (0.66) | - | 742.8 (112.6) | - | 0.25 (0.11) | - |
| N14 | 132.3 (50.2) | - | 0.48 (0.14) | - | 1957.9 (317.1) | - | 0.23 (0.07) | - |
| N28 | 7.8 (3.3) | - | 0.12 (0.15) | - | 438.5 (112.1) | - | 0.23 (0.15) | - |
| N + F | 130.3 (33.5) | - | 3.87 (0.04) | - | 1391.3 (424.6) | - | 0.13 (0.04) | - |
| N + F14 | 160.4 (44.5) | - | 3.39 (0.03) | - | 719.0 (126.4) | - | 0.07 (0.03) | - |
| N + F28 | 69.6 (17.9) | - | 0.61 (0.16) | - | 956.5 (152.07) | - | 0.19 (0.04) | - |
| F + N14 | 105.2 (23.1) | - | 6.94 (2.65) | - | 1793.9 (452.2) | - | 0.28 (0.10) | - |
| F + N28 | 46.1 (22.7) | - | 0.21 (0.10) | - | 520.3 (68.0) | - | 0.04 (0.02) | - |
| F14 + N14 | 68.4 (16.8) | - | 1.57 (0.52) | - | 1479.0 (224.8) | - | 0.39 (0.11) | - |
| F14 + N28 | 14.5 (7.4) | - | 0.34 (0.17) | - | 960.8 (316.7) | - | 0.15 (0.06) | - |
| N14 + F28 | 60.3 (12.0) | - | 2.50 (1.71) | - | 973.6 (265.7) | - | 0.27 (0.10) | - |
| N28 + F28 | 4.6 (2.9) | - | 0.29 (0.17) | - | 421.9 (147.1) | - | 0.09 (0.05) | - |

**Table 6.** Average number (95% c.i.) of nematode damage on stems and tubers and number of nematodes in each plant part and potting medium in treatments with addition of the root-lesion nematode *Pratylenchus penetrans* (Pp) in experiment 2. Treatments with the same letter within a column and experiment are not significantly different at a level of $p < 0.05$.

| Treatment | Stem Lesion | No. of P pg$^{-1}$ Stem | Tuber Lesion | Probability (%) Tuber Lesion | No. of Pp g$^{-1}$ Tuber | No. of Pp g$^{-1}$ Root | No. of Pp g$^{-1}$ Potting Medium |
|---|---|---|---|---|---|---|---|
| N, N + F, N + F14, N+ F28 | 11.3 (9.2–13.9) a | 103 (60–159) a | 5.0 (2.6–9.9) a | 34 (24–45) a | 1.8 (0.9–2.9) a | 800 (631–1016) b | 0.11 (0.05–0.19) ab |
| N14, N14 + F, N14 + F14, N14 + F28 | 12.3 (10.1–15.1) a | 76 (40–124) a | 7.6 (3.4–16.8) a | 39 (29–50) a | 1.6 (0.8–2.6) a | 1244 (927–1670) a | 0.21 (0.12–0.32) a |
| N28, N28 + F, N28 + F14, N28 + F28 | 3.8 (3.0–4.8) b | 6 (0.1–24) b | 1.3 (0.6–2.6) b | 12 (7–19) b | 0.1 (0.0–0.5) b | 396 (264–595) c | 0.05 (0.01–0.11) b |

In experiment 1, the number of fungal-feeding nematodes differed among the treatments ($p = 0.013$), with more fungal-feeding nematodes in the stem in the treatment with nematodes at planting and fungal addition after 14 days ($p < 0.001$) and fungus at planting and nematode addition after 28 days ($p = 0.009$) compared with the treatments with only nematodes (Table 5). There were also more fungal-feeding nematodes in these two combination treatments compared with the treatment with nematodes at planting and fungus added after 28 days ($p = 0.002$ and $p = 0.044$, respectively).

### 3.2. Impact on Stolons

In experiment 1, the dry weight of stolons was significantly affected by treatment with the fungus reducing the biomass regardless of time of inoculation ($p < 0.001$; Table 2), apart from when nematodes were added at start and the fungus at 28 days, which was not significantly different from the control and nematode only treatment. In the second experiment, the dry weight of the stolons was reduced by fungal addition at the start and at 14 days ($p < 0.001$; Table 2). The fungus produced sclerotia on the stolons, which were more numerous when added late and the number of necrotic lesions was also affected by time of fungal inoculation (Table 7).

**Table 7.** Dry weights (g; mean (SE)) and numbers of different parts of potato plants subjected to the plant pathogenic fungus *Rhizoctonia solani* (F) and the root-lesion nematode *Pratylenchus penetrans* (N) in experiment 2. Treatments with the same letter within a column and experiment are not significantly different at a level of $p < 0.05$.

| Treatment | Stolons Nectrotic Lesions | No. of Sclerotia on Stolons | Probability (%) Elephant Hide | Probability (%) Black Scurf | No. of Pp $g^{-1}$ Tuber |
|---|---|---|---|---|---|
| C | 0 | 0 | 0 | 0 | 0 |
| N, N14, N28 | 23.4 (14.9–37.0) a | 0 | 0 | 0 | 0.4 (0.0–1.0) b |
| F, F + N, F + N14, F + N28 | 7.3 (4.5–11.8) b | 1.9 (1.2–3.2) b | 42 (31–55) b | 58 (43–72) b | 2.0 (1.0–3.3) a |
| F14, F14 + N, F14 + N14, F14 + N28 | 5.8 (3.5–9.5) b | 2.4 (1.5–3.9) b | 76 (65–84) a | 80 (68–89) a | 1.2 (0.5–2.3) ab |
| F28, F28 + N, F28 + N14, F28 + N28 | 13.7 (8.6–21.8) a | 7.6 (4.9–11.7) a | 69 (57–80) a | 87 (76–93) a | 0.5 (0.1–1.3) b |

### 3.3. Impact on Tubers

The dry weight of the tubers was not affected by the fungus and nematodes in experiment 1 ($p = 0.057$, Figure 1). In experiment 2, the dry weight of tubers was reduced by both fungus and nematodes, regardless of the time of inoculation (Figure 2). The combination of fungus and nematodes reduced on average the yield by 11.7 g ($p < 0.001$) compared with the control, 6.6 g compared with the average of the fungus only treatments ($p < 0.001$), and 2.8 g compared with the nematode only treatments ($p = 0.023$). Addition of fungus at planting and nematodes at 28 days resulted in a higher dry weight than addition of both nematodes and fungus at planting ($p = 0.022$) and addition of fungus at planting and nematodes after 14 days ($p = 0.023$). Addition of fungus and nematodes had no significant effect on the number of tubers in either of the experiments.

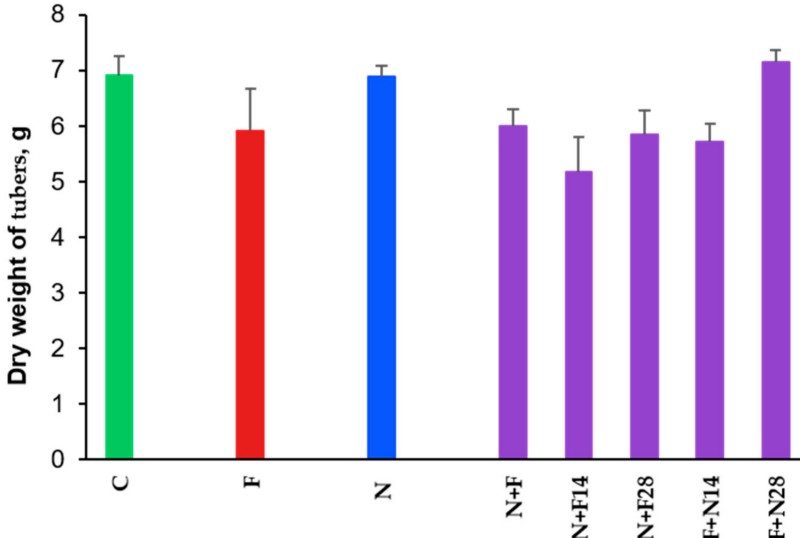

**Figure 1.** Dry weights (g; mean (SE)) of potato tubers subjected to different combinations of the plant pathogenic fungus *Rhizoctonia solani* (F) and full nematode community (N) in experiment 1 compared with the control (C, no addition).

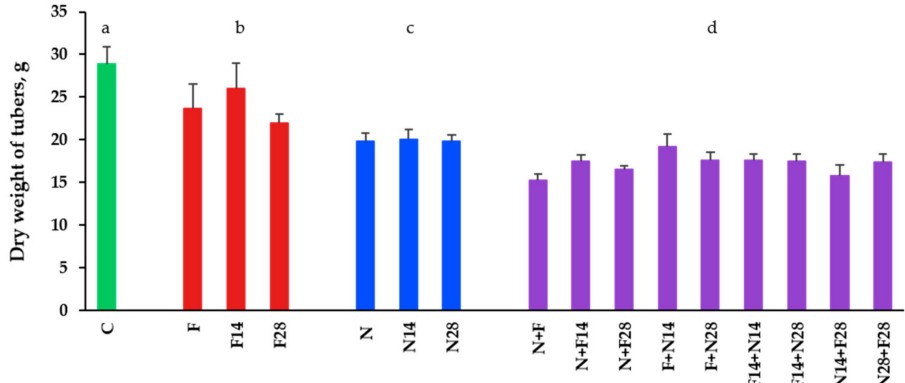

**Figure 2.** Dry weights (g; mean (SE)) of potato tubers subjected to different combinations of the plant pathogenic fungus *Rhizoctonia solani* (F) and the root-lesion nematode *Pratylenchus penetrans* (N) in experiment 2. The treatments were grouped into control (C, no addition), fungus only, nematode only, and the combinations. All groups of treatments were significantly different at a level of $p < 0.05$ as visualised by different letters above the bars.

The time of fungal inoculation influenced the amount of black scurf and elephant hide on the tubers. The probability of both black scurf and elephant hide was higher when the fungus was added at 28 days compared with when the fungus was added at planting in both experiments (Table 3, Table 4, and Table 7). In the second experiment, the odds increased for elephant hide and black scurf if the fungus was added at 14 days as well. Nematode damage was visible on the tubers in experiment 2, while symptoms of nematode feeding were not observed on the tubers in experiment 1. Addition of fungus did not interact with nematodes regarding symptoms, but the time of nematode inoculation was important for the amount of nematode damage and number of nematodes in the tubers (Table 6). The amount of nematode damage was significantly lower when nematodes were added at 28 days and the probability of nematode damage on tubers was highest when nematodes were added at 14 days.

Although there was no visible damage of nematodes in experiment 1, the tubers did contain plant-parasitic nematodes in both experiments (Table 5). The number of plant-parasitic nematodes in the tubers was higher when the nematodes were added at the start and at 14 days in experiment 2 (Table 6). Fungal addition at 28 days resulted in lower numbers of plant-parasitic nematodes in all nematode

treatments, compared with addition at the start in experiment 2 ($p$ = 0.014, Table 7). The number of nematodes in tubers was not different between the treatments in experiment 1 ($p$ = 0.09) (Table 5).

### 3.4. Impact on Roots

The dry weight of roots was not significantly affected by treatment in experiment 1 ($p$ = 0.10). Addition of nematodes reduced the root dry weight ($p$ < 0.001) in experiment 2 and there was an interaction between addition of nematode and fungus ($p$ = 0.047) (Table 2). The roots were always covered with sclerotia if the fungus had been added and had brown lesions or sections regardless of treatment and experiment. Plant-parasitic nematodes were abundant in the roots, especially in the second experiment, where the plants that were inoculated at 14 days had the highest abundance (Tables 5 and 7). There were, however, no significant differences among the treatments in experiment 1. On the other hand, the number of fungal-feeding nematodes differed among the treatments in experiment 1 ($p$ = 0.013) with more fungal-feeding nematodes when the fungus was added at planting and nematodes after 14 days compared with three other nematode treatments (Table 5).

### 3.5. Potting Medium

Both plant-parasitic and fungal-feeding nematodes were found in the potting medium in experiment 1, but neither group was significantly affected by the different treatments (Table 5). In experiment 2, more nematodes were found in the potting medium when inoculated after 14 days and the lowest amount was found after the latest addition (Table 6).

## 4. Discussion

Addition of root-lesion nematodes and *R. solani* affected the potato plants in some way for almost all treatments. Two of our hypotheses were confirmed as we found that the tuber yield was affected by the combination of nematodes and *R. solani* and that the time of inoculation influenced the severity of both fungal and nematode damage. However, contrary to our hypotheses, we found that the quality of the tubers and the fungal damage on the plant were not significantly dependent on the presence of nematodes prior to fungal inoculation (hypothesis no. 5 not confirmed). The severity of stem canker did not increase in the presence of nematodes and the stem canker was even less severe when nematodes were added in combination with the fungus, whereof hypothesis no. 1 was not confirmed.

In both experiments, the interaction between the two pathogenic organisms was most substantial regarding the tubers. The combination of plant-parasitic nematodes and *R. solani* resulted in lower tuber yield in both experiments, which was particularly observed in the second experiment, but the interaction was not synergistic. The root-lesion nematode *P. penetrans* may alone cause considerable yield loss, by 30% to 70%, in the potato crop, mainly through impact on the roots [28–30]. The reduction in root biomass restricts the optimal uptake of water and nutrients, which is needed for adequate potato yield. Therefore, the yield loss found in our experiments most likely depended on the observed reduction of root biomass in the presence of nematodes, regardless of the time of inoculation (Table 2, Figure 1, and Figure 2). In addition, in some treatments, there was an interaction effect of nematodes and fungus on the root biomass, and similar results were also observed in a previous study, where the potato cultivar Kuras was inoculated with a full nematode community dominated by root-lesion nematodes in combination with *R. solani* [12].

Contrary to our first hypothesis, plant-parasitic nematodes reduced the number of stem canker lesions on the stems instead of making them more severe, and the nematodes did not affect fungal damage on stolons or tubers either, as hypothesized. One possible explanation may be that the nematodes activated resistance mechanisms in the potato plants in the same way as root-knot nematodes may induce defence mechanisms in tomato [31].

In accordance with our fourth hypothesis, the time of inoculation influenced fungal and nematode damage. Fungal skin tuber damage increased if the fungus was added at 14 and 28 days, which may be because of infiltration of the added mycelium directly on the developing tubers. The fungal damage on

the stolons was also affected by the time of fungal addition with increased necrosis if the fungus was added early and late, which may be because of the prolonged time of exposure when the mycelium was present during growth and that the mycelium could infiltrate directly on the stolons when added late, respectively.

Regarding nematode damage, the earlier the nematodes had the opportunity to colonise the plantlet, the more severe damage they achieved. The nematode damage and the number of nematodes in the plant parts were higher if the nematodes were added early, denoting that the nematodes thrived on the potato plants, even though not shown as visual damage in the first experiment. The necrotic lesions on tubers and stems, as well as the number of nematodes in the plant parts, were fewer when the nematodes were added late (at 28 days). The number of nematodes also increased in the tubers when the fungus was added early in experiment 2. Surprisingly, root-lesion nematodes were extracted from the stems. These nematodes generally do not occur in stems, but possibly did here because of the high competition for food in the limited soil volume. The stems may, however, be more difficult to feed from when they grow older, hence the lower abundance when nematodes were added after 28 days. Nematodes may first attack the roots and then later attack the stem when the fungus has affected the stem tissue [32].

The time of inoculation did also influence the number of plant-parasitic nematodes in the potting material, because the highest abundance was found when added after two weeks and the lowest abundance was found after addition at day 28 (experiment 2). This lower abundance in the potting material coincides with the higher abundances in the plant parts and indicates that the nematodes must have entered and left the plant parts regularly. *Pratylenchus penetrans* have a short life cycle of 4–8 weeks and the females can produce thousands of nematodes in the roots at once, which may explain the high amount of nematodes found at harvest [33].

Although no damage of nematodes was visible in the first experiment, the tubers did contain both plant-parasitic and fungal-feeding nematodes. The number of plant-parasitic nematodes in the tubers differed among the treatments, but there were differences between the two experiments, whereof no general conclusions about the time of inoculation and order of appearance could be drawn. Increased numbers of plant-parasitic nematodes in the tubers were observed for the combination treatment in five of six cultivars in a previous experiment [12].

The anastomosis group of the isolate used was not the one that usually affects potato (AG3), but AG2-1 is also pathogenic on potato and can be more aggressive on shoots and stems, and produces many small canker lesions [34,35]. The anastomosis group occurs in Swedish and Finnish agricultural soils, but was reported as less aggressive, and to at least form less sclerotia on the tubers [36,37]. The fungal strain used in these experiments did, however, demonstrate good efficacy in producing both stem canker and sclerotia on the tubers.

## 5. Conclusions

It is essential to understand and appreciate the importance of relationships between pathogens in order to control disease through appropriate management methods; thereof, more research is needed to unravel these questions. Our main conclusions are that the yield is affected by co-occurrence of root-lesion nematodes and *R. solani*, and that the nematodes may interact with the potato plant, leading to less stem canker. Our results highlight the importance of analysing the presence of nematodes in the field to be able to create protective strategies for an efficient potato production. Crop rotation with at least four potato free years and other agricultural measurements may be useful to reduce the population of *R. solani* in the field [1,38], and the effect of the crops grown in the potato free period on root-lesion nematodes needs to be taken into consideration.

**Author Contributions:** Conceptualization, E.E. and M.V.; methodology, E.E., M.G., and M.V.; formal analysis, M.F.A., A.F., J.E.E., and J.K.; investigation, E.E., M.G., and M.V.; writing—original draft preparation, E.E., M.G., and M.V.; writing—review and editing, E.E., M.G., M.F.A., A.F., J.E.E., J.K., and M.V.; visualization, E.E.; supervision, E.E. and M.V.; project administration, M.V.; funding acquisition, M.V.

**Funding:** This research was funded by The Swedish Farmers' Foundation for Agricultural Research (Stiftelsen Lantbruksforskning SLF), grant number H1142045.

**Acknowledgments:** We thank personnel and students at the Department of Ecology, SLU, for help during the harvest of the experiments.

**Conflicts of Interest:** The authors declare no conflict of interest. The funders had no role in the design of the study; in the collection, analyses, or interpretation of data; in the writing of the manuscript; or in the decision to publish the results.

## Appendix A

**Table A1.** Grading of plant parts in experiment 1 and the severity of stem canker and "elephant hide" were graded in the same way in experiment 2.

| Plant Parts | Nematode Damage | | Sclerotia | | Stem Canker | |
|---|---|---|---|---|---|---|
| **Stem** | 0 | No damage | 0 | No sclerotium | 0 | No symptoms |
| | 1 | 1–10 stripes | 1 | 1–10 sclerotia | 1 | Small lesions |
| | 2 | 11–20 stripes | 2 | 11–20 sclerotia | 2 | Large lesions |
| | 3 | >20 stripes | 3 | >20 sclerotia | 3 | Lesion surrounds stem |
| | | | | | 4 | Dead * |
| **Stolon** | **Lesions** | | **Sclerotia** | | **% Brown Stolons** | |
| | 0 | No damage | 0 | No sclerotium | 0 | No symptoms |
| | 1 | 1–10 stripes | 1 | 1–10 sclerotia | 1 | Small lesions |
| | 2 | 11–20 stripes | 2 | 11–20 sclerotia | 2 | Large lesions |
| | 3 | >20 stripes | 3 | >20 sclerotia | 3 | Lesion surrounds stem |
| **Tubers** | **Nematode Damage** | | **Sclerotia** | | **"Elephant Hide"** | |
| | 0 | No damage | 0 | No sclerotium | 0 | No symptoms |
| | 1 | 1–10 spots | 1 | 1–10 sclerotia | 1 | Few small |
| | 2 | 11–20 spots | 2 | 11–20 sclerotia | 2 | Cover more than ¼ |
| | 3 | >20 spots | 3 | >20 sclerotia | 3 | Cover more than ½ |
| **Roots** | **Nematode Damage** | | **Sclerotia** | | **"% Brown Roots"** | |
| | 0 | No damage | 0 | No sclerotium | 0 | 0–9 |
| | 1 | Few stripes | 1 | 1–10 sclerotia | 1 | 10–24 |
| | 2 | Stripes on several roots | 2 | 11–20 sclerotia | 2 | 25–49 |
| | 3 | Stripes on most roots | 3 | >20 sclerotia | 3 | 50–100 |

* Those stems that were grades as a four ("dead") were often short and still submerged.

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
