# Peer review of "Temporal Interactions between Root-Lesion Nematodes and the Fungus Rhizoctonia Solani Lead to Reduced Potato Yield"

_agronomy, doi:10.3390/agronomy9070361_

Reviewer 1 Report

Dear Authors,

Re: Manuscript ID agronomy-530977

Temporal interactions between free-living plant-parasitic nematodes and the fungus Rhizoctonia solani lead to reduced potato yield

Nematode-fungus disease complexes have serious implications for a range of crops world-wide including potatoes. This manuscript builds on previously published work by researchers from SLU by considering interactions between free living PPN and Rhizoctonia solani under glasshouse conditions. The two experiments are specifically investigating the temporal occurrence/introduction of nematodes, principally Pratylenchus penetrans, and R. solani in potatoes. Overall, the manuscript is well presented and relatively easy to follow. The experiments and data analysis appear to have been conducted appropriately.

Suggested corrections: -

Line 25 (and elsewhere) - There are numerous instances where damage is written as damages. Damage can be singular and plural.

Line 89 - Awkward/cumbersome sentence. Other suggestions 'Based on preliminary results from......'  With reference to observations from...'

Line 93 - Which sub-group of AG2? I am presuming that you used AG 2-1

Line 104 - Provide a reference or more detail for this method (maize culture).

Line 112 -......one filter paper was used to cover the bottom....

Line 164 - Given the experimental design, I am quite surprised that factorial analysis was not conducted for experiment 2. This would have been very interesting for examining interactions.

Line 282 - Text on the results does not follow a logical order. Report on experiment 1 before moving on to experiment 2.

Line 284 - It is standard practice to report P values to 0.001. Correct here and elsewhere.

Line 285 Figure 2?

Line 286 - 'The same pattern was observed'. Looking at your figures, I do not concur with this sweeping statement.

Line 286 - Figure 1

Line 297 - Inconsistency between presentation/post hoc analysis. Figure 1 shoes detail for treatments (preferred) whereas Figure 2 groups independent and combined treatments.

Line 317 - Typo (Impact)

Line 333 - Please consider discussing opportunities for further work. For example, dose (fungal inoculum and nematode densities) and anastomosis group may be interesting topics.

Line 345 - Change 'specially' to 'particularly'

Line 363 - Delete 'numbers of'

Line 374 - Awkward sentence. Revise.

Lines 377-8 - I have never heard about root lesion nematodes invading stem tissue before. End of sentence from '...and have destroyed..' needs revision

Line 404 - Reference needed for sentence on crop rotation.

Line 406 - ....may be more difficult to manage...In any case, you are generalizing here! Be more specific.

Author Response

Line 25 (and elsewhere) - There are numerous instances where damage is written as damages. Damage can be singular and plural.

Reply: Thank you for the information, we have change accordingly.

Line 89 - Awkward/cumbersome sentence. Other suggestions 'Based on preliminary results from......'  With reference to observations from...'

Reply: Thank you for the suggestion and we changed to “With reference to observations from experiment 1 and the experimental set up we performed a repetition of the experiment…”

Line 93 - Which sub-group of AG2? I am presuming that you used AG 2-1

Reply: Yes, it was. Thank you for noticing

Line 104 - Provide a reference or more detail for this method (maize culture).

Reply: we added information in the manuscript: …” The maize plants were grown in sterilized sand and new seeds were sown after the senescence of the plants, which were cut and the roots was left in the pots as feed for the nematodes.”

Line 112 -......one filter paper was used to cover the bottom.... 

Reply: we have changed the sentence as suggested.

Line 164 - Given the experimental design, I am quite surprised that factorial analysis was not conducted for experiment 2. This would have been very interesting for examining interactions.

Reply to Line 164 and 297: For experiment 2 we opted for a model selection approach, rather than estimating the full-factorial model. Model selection was done through backwards-elimination, starting in the full-factorial model. The background to this choice is

i) that the model selection approach includes a test of the factorial model, as it is the starting model,

ii) that the approach allows testing an effect of addition/non-addition for experiments where there is no effect of time of addition, and

iii) that the approach gives higher statistical power for tests of the main effects.

Line 282 - Text on the results does not follow a logical order. Report on experiment 1 before moving on to experiment 2.

Reply to comments on line 282, 285 and 286: We have changed the order of presentation and revised the sentences.

Line 284 - It is standard practice to report P values to 0.001. Correct here and elsewhere.

Reply: thank you for noticing. We have changed accordingly.

Line 285 Figure 2?

Line 286 - 'The same pattern was observed'. Looking at your figures, I do not concur with this sweeping statement.

Line 286 - Figure 1

Line 297 - Inconsistency between presentation/post hoc analysis. Figure 1 shoes detail for treatments (preferred) whereas Figure 2 groups independent and combined treatments.

Reply: see comment for Line 164

Line 317 - Typo (Impact)

Reply to Line 317, 345 and 363: Thank you for noticing and we have made the suggested changes.

Line 333 - Please consider discussing opportunities for further work. For example, dose (fungal inoculum and nematode densities) and anastomosis group may be interesting topics.

Reply: thank you for the suggestions. We have added some future aspects.

Line 345 - Change 'specially' to 'particularly' 

Line 363 - Delete 'numbers of'

Line 374 - Awkward sentence. Revise.

Reply: we have removed the sentence.

Lines 377-8 - I have never heard about root lesion nematodes invading stem tissue before. End of sentence from '...and have destroyed..' needs revision

Reply: Yes, we were surprised as well but the stems were rinsed so there should not the any nematodes that was attached to the stem surface. We have revised the sentence. To Surprisingly, root-lesion nematodes were extracted from the stems. These nematodes generally do not occur in stems but possibly did that here due to high competition for food in the limited soil volume.

Line 404 - Reference needed for sentence on crop rotation.

Reply: inserted two

Line 406 - ....may be more difficult to manage...In any case, you are generalizing here! Be more specific.

Reply: we have revised the paragraph.

Reviewer 2 Report

Diverse pathogens are a severe problem in potato production, which may interact and synergistically affect the crop. The authors investigated the interactive effects of R. solani and root-lesion nematodes on potato plants, especially the damage on tubers. Both pathogens caused severe damage, while the effects were not synergistic. In contrast to our expectations, prior inoculation of the nematodes did not assist R. solani infecting the plant. Stem canker was even reduced by the nematodes, presumably by activation of defense mechanisms. The plant age influenced the damage by the pathogens. The study was well designed. The results are interesting for nematologists and agronomists. Presentation of hypotheses and results is concise and clear. The title may not well reflect the main findings. The usage of the term "free-living nematodes" is irritating to me. I think it is mostly used as a contrast to endoparasitic nematodes. Maybe the term could be replaced by "root-lesion nematodes", just adding the explanation that in experiment 1 also other nematodes were present?

Minor comments:

Clearly state in the abstract, that there was no synergistic interaction of both pathogens, and that Pratylenchus did not assist R. solani infecting the plant.

L. 25 skip "For example,"

L. 129 on average

L. 142 ...inoculation. This was...

L. 146 In experiment 1, the plants...

L. 228 one bracket too much

L. 317 Impact

L. 320 ...was not significantly affected by...

L. 339 dependent on

L. 352 results were

Author Response

Clearly state in the abstract, that there was no synergistic interaction of both pathogens, and that Pratylenchus did not assist R. solani infecting the plant.

Reply: we have revised the section and added the suggested changes. The suggestions below were all made in the revision. Thank you for noticing the errors.

L. 25 skip "For example,"

L. 129 on average                                                       

L. 142 ...inoculation. This was...

L. 146 In experiment 1, the plants...

L. 228 one bracket too much

L. 317 Impact

L. 320 ...was not significantly affected by...

L. 339 dependent on

L. 352 results were